# Pro-Inflammatory Microenvironment Modulates the Transfer of Mutated TP53 Mediated by Tumor Exosomes

**DOI:** 10.3390/ijms22126258

**Published:** 2021-06-10

**Authors:** Rossana Domenis, Adriana Cifù, Catia Mio, Martina Fabris, Francesco Curcio

**Affiliations:** 1Department of Medical Area, University of Udine, 33100 Udine, Italy; adriana.cifu@uniud.it (A.C.); catia.mio@uniud.it (C.M.); francesco.curcio@uniud.it (F.C.); 2Institute of Clinical Pathology, Azienda Sanitaria Universitaria Integrata Friuli Centrale, 33100 Udine, Italy; martina.fabris@asufc.sanita.fvg.it

**Keywords:** exosomal DNA, pro-inflammatory cytokines, oncogene transfer

## Abstract

Exosomes released from tumor cells are instrumental in shaping the local tumor microenvironment to allow cancer progression. Recently, it has been shown that tumor exosomes carry large fragments of dsDNA, which may reflect the mutational status of parental cells. Although it has been described that a stressful microenvironment can influence exosomal cargo, the effects on DNA packing and its transfer into recipient cells have yet to be investigated. Here, we report that exosomes derived from SW480 (human colorectal adenocarcinoma cell line) cells can carry dsDNA fragments containing the entire coding sequence of both *TP53* and *KRAS* genes, harboring the SW480-related *TP53* c.818G > A and *KRAS* c.35G > T typical mutations. We also report the following: that cell stimulation with lipopolysaccharides (LPS) promotes the selective packaging of the *TP53* gene, but not the *KRAS* gene; that exosomes secreted by SW480 cells efficiently transfer the mutated sequences into normal CCD841-CoN colon epithelial and THLE-2 hepatic cells; that this mechanism is more efficient when the cells had been previously incubated with pro-inflammatory cytokines; that the *TP53* gene appears actively transcribed in both recipient cells; and that mutated mRNA levels are not influenced by cytokine treatment. Our data strongly suggest that pro-inflammatory stimulation promotes the horizontal transfer of an oncogene by exosomes, although this remains a rare event. Further studies are needed to assess the impact of the oncogenic transfer by exosomes in malignant transformation and its role in tumor progression.

## 1. Introduction

Exosomes are nanometer-sized active carriers with a key role in the intercellular communication system. They are involved in several physiological functions and pathological conditions [1]. Since their discovery, numerous studies have been conducted to understand their molecular composition and biological function.

Exosomes transport diverse and specific repertoires of nucleic acids and proteins, which suggests the existence of a sorting mechanism that regulates the composition of their cargo [2]. Recently, in patients with pancreatic cancer, large fragments of double-stranded genomic DNA have been identified in circulating exosomes [3], suggesting that a large proportion of plasma cell-free human DNA is transported in exosomes [4] and that this mechanism is important for maintaining cellular homeostasis by allowing the elimination of damaged DNA [5]. Indeed, the contents of micronuclei, formed as a consequence of genomic instability, are transferred into multivesicular bodies via tetraspanin proteins and then loaded into exosomes [6]. The inhibition of exosome secretion leads to the accumulation of nuclear DNA in the cytoplasm and triggers a reactive oxygen species (ROS)-dependent DNA damage response (DDR) [5], which in turn results in cell cycle arrest and apoptosis.

It has been shown that exosomal DNA can be used as a diagnostic marker in several pathologies [7], including tumor malignancies [8,9] and neurological disorders [10]. Tumor-derived microvesicles show higher DNA contents than those derived from normal cells [11,12]. Furthermore, the finding that the DNA contained in exosomes reflects the mutational status of parental tumor cells [3,9,13,14] highly supports their potential utility as a non-invasive circulating biomarker that can be used to facilitate early cancer diagnosis and the monitoring of treatment response [9]. In addition, it has also been shown that the extracellular vesicles (EVs) produced by tumor cells can transfer their genetic cargo to recipient cells [15]. Whether or not this has implications for the phenotype of the recipient cells is yet to be clarified; for instance, the transfer of mitochondrial mtDNA by EVs has been associated with the increased self-renewal potential of cancer stem-like cells and may be a mechanism contributing to the onset of tumor resistance to hormonal therapy [16].

Using a whole-genome sequencing (WGS) approach, it has been shown that large fragments of both single- and double-stranded DNA are present in EVs, covering the entire human genome [3]; furthermore, their ability to package the whole coding region of selected genes ensures their potential transcription within recipient cells [17]. Indeed, exosomal DNA could be transferred into the nucleus and transcribed by endogenous transcription factors [18,19], resulting in the transformation of the surrounding normal cell, similar to an infectious process. Fischer et al. demonstrated that exogenous DNA transfected into bone marrow mesenchymal stromal cells can be transferred into wild-type cells by exosomes and propagated, probably in an integrated state, in the host genome. Although this has been classified as a very rare event, transfer has been unambiguously detected [15].

Although there is evidence of the transforming capacity of exosomes, this process has not yet been fully elucidated, especially with regard to its regulatory mechanisms. Changes associated with the tumor microenvironment, such as hypoxia, acidification, and inflammation, may activate cellular stress-response mechanisms and increase exosome production, while also affecting their content [20,21,22]. Recently, we have shown that the activation of toll-like receptor 4 (TLR4) of tumor cells induces inflammatory signaling, which influences the composition of exosomes and boosts their immunosuppressive capacity [23].

To date, to the best of our knowledge the role of the microenvironment in the horizontal transfer of DNA carried by tumor exosomes has not been studied. Our aim in this study is to evaluate the hypothesis that a pro-inflammatory stimulus may modulate the transfer of mutated genes into exosomes and support their transcription into recipient cells, thus amplifying the ability of exosomes to induce malignant transformation.

## 2. Results

### 2.1. Detection of TP53 c.818G > A and KRAS c.35G > T Mutations in Exosomal DNA of SW480 Cells

Exosomal DNA was extracted from vesicles isolated from the supernatants of SW480 cells upon lipopolysaccharides (LPS) stimulation, as previously described [23]. As shown in Figure 1A, the activation of cells with LPS did not affect the amount of DNA packed into exosomes, whereas the average size of DNA fragments was slightly, but significantly, higher in exosomes released after LPS treatment (3121.8 ± 270.3 bp vs 3718.5 ± 513.9 bp) (Figure 1B). The presence of *TP53* c.818G > A and *KRAS* c.35G > T mutations was investigated by droplet digital PCR using genomic DNA isolated by SW480 cells as a positive control. The amount of *TP53*-mutated copies was higher in exosomes produced from LPS-treated cells (Figure 1C), whereas no difference was observed for KRAS (Figure 1D).

To evaluate whether exosomes carried long DNA fragments with the full coding sequence of *TP53* and *KRAS*, next-generation sequencing (NGS) was performed using the Ion Torrent system. The graphical visualization of aligned reads with Tablet revealed that the full-length *TP53* and *KRAS* genes were included (Figure 2).

### 2.2. Pro-Inflammatory Cytokines Potentiate Oncogenic DNA Transfer by Exosomes in Normal Colon and Liver Epithelial Cells

To assess whether oncogenic DNA sequences were transferred into recipient cells by exosomes, CCD841-CoN normal epithelial cells were treated with SW480-derived exosomes every other day, and the DNA content was evaluated 24 h after the last treatment. In addition, to assess the influence of the pro-inflammatory microenvironment, recipient cells were stimulated with IL1β, IL6, and TNFα cytokines.

We observed that the exosomes were capable of transferring oncogenic DNA into CCD841-CoN cells, and that this phenomenon was more evident after incubating the cells with pro-inflammatory cytokines, as demonstrated by the significant increase in copies of mutated *TP53* in the DNA extracted from cells after treatment (Figure 3A). We also observed that this phenomenon is maximal when exosomes derived from LPS-treated SW480 cells are used. Similar results were also observed with *KRAS* (Figure 3B). We hypothesize that the LPS treatment of SW480 affects the composition of released exosomes, enriching their content of certain oncogenes (as is the case for *TP53*) and also promoting their receptor- and raft-mediated endocytosis by recipient cells (as is likely the case for *KRAS* in our experimental model)

The presence of mutated DNA was also assessed in CCD841-CoN recipient cells cultured for an additional 10 days without exosomes. Mutated *TP53* was detected in DNA extracted from CCD841-CoN cells, suggesting its possible integration into their genome (Figure 3A). The presence of mutated *TP53* was detected for up to 30 days without exosome treatments (data not shown) and, because no significant differences were observed compared with the analysis after 10 days, the experiment was not extended beyond that.

Interestingly, the same phenomenon was not observed for mutated *KRAS*, indicating that this, although internalized in recipient cells, was not integrated into their genome.

To confirm that mutated *TP53* was integrated and actively transcribed, cellular RNA was extracted and analyzed by droplet digital PCR. The results confirmed the presence of mutated *TP53* mRNA in cytokine-treated cells, although no difference was observed between exoCTRL and exoLPS treatment (Figure 3C).

Next, we evaluated whether SW480-derived exosomes were able to mediate the horizontal transfer of oncogenes into cells from a different tissue. Specifically, to simulate the effect on cells belonging to a possible metastatic site, we used immortalized normal hepatocytes as recipient cells. Similarly to epithelial cells, we found mutated TP53 in the DNA extracted from THLE-2 cells treated with SW480-derived exosomes, and the amount of copies increased in cells grown under pro-inflammatory conditions, although no significant difference was observed between the control and LPS-derived exosomes (Figure 4A). It is worth mentioning that mutated *KRAS* was undetectable in liver recipient cells (data not shown). As with CCD841-CoN, the presence of mutated DNA was also assessed in the THLE-2 recipient cells cultured for an additional 10 days without exosomes. Again, our results showed that mutated *TP53* was integrated into the genome of the recipient cells, as we also found both mutated DNA (Figure 4A) and mRNA (Figure 4B) in cells cultured without exosomes for 10 days. No differences were observed in the different conditions analyzed.

## 3. Discussion

Exosomes play a key role in cell-to-cell communication and can modulate biological functions through the transfer of bioactive molecules. The role of exosomes in tumorigenesis and cancer progression has been intensively investigated [24], and the scientific community agrees on the ability of tumor cells to use exosomes to shape the local or distal microenvironment to their own advantage and to promote tumor expansion. The discovery that tumor exosomes can carry large fragments of dsDNA containing mutated sequences [18] has highlighted the possibility that vesicles may transfer oncogenes and may contribute to malignant transformation. To date, although several studies have reported that exosomal DNA could be transferred horizontally between cells, it is still unclear whether the mutated sequences can be integrated into the genome of the recipient cells. In support of this hypothesis, it has been suggested that tumor-derived EVs contain genome-modulating retrotransposons [11], which could be transferred to normal cells.

Many studies have shown that genes transferred by exosomes into recipient cells can be transcribed [15,25], but the related oncogenic transformation seems to be transient [17,26]. On the other hand, the pathological effect of oncogene transfer by EVs was demonstrated by an in vivo study showing that the BCR/ABL fusion gene transferred by exosomes derived from K562 cells induces chronic myeloid leukemia in immunodeficient mice [27]. The uptake of apoptotic bodies derived from H-rasV12- and c-myc-transfected cells resulted in both the loss of contact inhibition and anchorage independence, as well as tumorigenicity in SCID mice [28]. Further studies focused on evaluating the transforming capacity of exosomes are needed.

Considering that the production rate and composition of exosomes have been profoundly affected by stress conditions [21], we hypothesize that the microenvironment plays a role by also influencing the mechanism involved in DNA loading into exosomes and its integration into recipient cells.

We observed that the activation of TLR4 on tumor cells by LPS promote the packaging of mutated *TP53* in exosomes. We reported an increase in the *TP53* copy number in LPS-derived exosomes, suggesting that the mechanism behind DNA sorting is not stochastic. Of note, it has recently been shown that functional KRAS protein transfer from cell to cell is an extremely rare event [29]. To date, the mechanisms by which nucleic acids are sorted and packaged within EVs have yet to be elucidated [30]. However, a growing number of publications have shown that the EVs derived from primary tumors can be preferably loaded with specific molecules [31,32], and that the expression of such molecules or lack thereof contributes to shaping the phenotype of the recipient cell.

Furthermore, we reported that treatment with pro-inflammatory cytokines resulted in an increase in mutated *TP53* within recipient cells, and its transcription in both colon epithelial and hepatic cells. We have to consider that, although the integration of mutated sequences transferred via EVs is a rare event, this mechanism may acquire more importance under conditions of continuous exosome release, as occurs in vivo.

Although no differences in mutated RNA levels were observed in the presence or absence of cytokine treatment, it is plausible to assume that a pro-inflammatory environment, by promoting exosome transfer, may increase the probability of oncogene integration. In support of this hypothesis, it has been shown that cancer-derived exosomes induce malignant transformation only in combination with a stimulus that promotes cell proliferation. In fact, a recent publication shows that exosomes released from pancreatic cancer-derived cell lines, unlike those produced by normal pancreatic cells, introduce mutations in NIH/3T3 recipient cells. Under proliferative stimulus, these cells are completely transformed and promote in in vitro culture the formation of foci whose cells can induce tumors when injected into mice [33].

In the susceptibility of cells chosen to mimic recipient cells adjacent or distal to the exosome-releasing tumor to integrate and express mutated DNA, we observed differences between colonic epithelial cells and hepatic cells. THLE-2 cells appeared to be less susceptible than colonic epithelial cells to the uptake of exosome-borne oncogenes. For example, *KRAS* p.Gly12Val was not detected in THLE-2 cells. Differences between cells have already been documented, demonstrating that normal cells are more resistant to the uptake of tumor-related EVs and thus to the transfer of oncogenes through them, while they become susceptible if they undergo malignant transformation [26].

In conclusion, our data show that pro-inflammatory stimulation promotes the horizontal transfer of the *TP53* and *KRAS* oncogenes by SW480-derived exosomes, and that the former is transcribed in recipient cells. Although the integration of the mutated gene remains to be considered as a rare event, it may be a relevant phenomenon in vivo. Further studies are needed to assess the impact of oncogene transfer by exosomes in malignant transformation and its role in tumor progression, especially under pro-inflammatory conditions.

## 4. Materials and Methods

### 4.1. Culture and Treatment of Cell Lines

The SW480 human tumor cell line was established from a primary colon adenocarcinoma and carries *KRAS* (NM_004985.5) c.35G > T (p.Gly12Val) and *TP53* (NM_000546.6) c.818G > A (p.Arg273His) mutations.

SW480 cells were cultured in Dulbecco’s modified essential medium (DMEM high glucose, Sigma-Aldrich, Darmstadt, Germany), supplemented with 10% FBS (Gibco, Waltham, MA USA) and 1% penicillin/streptomycin solution (Gibco) and treated with lipopolysaccharides (LPS) at a final concentration of 1 µg/mL (*E. coli* 055:B5 LPS, Sigma-Aldrich) for 24 h. It was washed three times with PBS and then the culture medium was replaced with a fresh medium, supplemented with 10% certified exosome-free serum (Gibco) [23]. After 24 h, the cell-conditioned medium was collected and stored at −20 °C until the exosome isolation step.

The CCD841-CoN normal human colonic epithelial cell line was cultured in Eagle’s Minimum Essential Medium, supplemented with 10% FBS (Gibco) and 1% penicillin/streptomycin solution (Gibco).

The normal human hepatocyte cell line THLE-2 was seeded on flasks pre-coated with 0.01 mg/mL fibronectin, 0.01 mg/mL bovine serum albumin (BSA) and 0.03 mg/mL collagen type I (all from Sigma-Aldrich) dissolved in growth medium. THLE-2 cells were cultured in BEGM medium (BEGM Bullet Kit, Lonza, Basel, Switzerland) without gentamicin/amphotericin and epinephrine supplementation, but with the addition of 5 ng/mL EGF (Peprotech, Rocky Hill, NJ, USA), 10% FBS (Gibco), and 1% penicillin/streptomycin solution (Gibco).

All cell lines were obtained directly from ATCC and cultured at 37 °C with 5% CO_2_.

### 4.2. Exosomal DNA Extraction and Analysis

Exosomes were isolated from the supernatants of SW480 cells by the polymer precipitation method (ExoQuick-TC, System Biosciences, Palo Alto, CA, USA) and quantified using the Exocet kit (System Biosciences), as previously described [26]. DNA was extracted from exosomes (3 × 10^10^) using a QIAamp DNA Mini Kit (Qiagen, Hilden, Germany), according to the manufacturer’s instructions, except that exosomes were lysed at 56 °C for 1 h [34]. DNA was eluted in 40 µL of AE buffer (10 mM Tris·Cl; 0.5 mM EDTA; pH 9.0), quantified using the high-sensitivity Qubit dsDNA assay kit (Thermo Fisher Scientific, Waltham, MA USA), and quality was analyzed using the Agilent DNA 12,000 reagent kit (Agilent, Santa Clara, CA, USA).

### 4.3. Library Preparation and Next-Generation Sequencing

Barcoded libraries were generated from 20 ng of DNA per sample using the Ion AmpliSeq HiFi mix (Ion AmpliSeq Library Kit Plus, Thermo Fisher Scientific) and two premixed pools of 952 primer pairs (Thermo Fisher Scientific), according to the manufacturer’s instructions. The clonal amplification of the libraries was performed by emulsion PCR on an Ion OneTouch™ 2 System. Sequencing was performed on the Ion Personal Genome Machine™ (PGM™) System using an Ion 318™ Chip Kit v2 BC and the Ion PGM™ Hi-Q™ View Sequencing Kit (all Thermo Fisher Scientific). Alignments were visualized with a Tablet v1.21.02.08 (https://ics.hutton.ac.uk/tablet/, accessed on 8 February 2021).

### 4.4. Data Analysis and Variant Prioritization

The Variant Caller v5.10 was used to process the data (Thermo Fisher Scientific). Annotation was performed with both Ion Reporter 5.10 (Thermo Fisher Scientific) and wANNOVAR (http://wannovar.wglab.org/, accessed on 2 February 2021), as previously described [35]. Germline variants were called when a position was covered at least 30×. Variant prioritization was based on the population frequency, quality values, and functional consequences. Variants were filtered based on their frequency in the European-descendent population (1000 Human Genomes Project, ESP6500SI, gnomAD, ExAC) and on clinical associations (NCBI dbSNP, ClinVar). Rare variants were defined as those with a minor allele frequency (MAF) <1%. Variants listed in ClinVar as “not-pathogenic”, “probable-not-pathogenic”, “drug response”, or “other” were excluded. Variants were classified according to the American College of Medical Genetics and Genomics (ACMG)/Association for Molecular Pathology (AMP) guidelines [36] on both the Varsome and wIntervar databases. After filtering and prioritization, exonic, splicing, stop-gain, stop-loss, and frameshift indel variants were retained for further evaluation.

### 4.5. Droplet Digital PCR (ddPCR)

To investigate the presence of *KRAS* c.35G > T and *TP53* c.818G > A mutations, 10 ng of exosomal DNA and validated primers and probe kits were used. Probes for wild-type sequences were 5′-labeled with 6-carboxyfluorescein (6-FAM), while probes for mutated sequences were 5′-labeled with hexachloro–fluorescein (HEX). All probes were 3′-labeled with black hole quencher 1 (BHQ-1).

Droplets were generated using the QX200™ droplet generator (Bio-Rad, Hercules, CA, USA), and PCR was performed on the T-100 thermal cycler (Bio-Rad) with the following protocol: 95 °C for 10 min, followed by 40 cycles at 94 °C for 30 s, 55 °C for 1 min, and 98 °C for 10 min (ramp rate 2 °C/s). Samples were analyzed using the QX200™ Droplet Reader (Bio-Rad) and droplet fluorescence data were analyzed using QuantaSoft™ version 1.7.4 (Bio-Rad). Discrimination between negative and positive droplets was achieved by manually setting a fluorescence amplitude threshold based on the signal from the parental cells used as a positive control.

### 4.6. Exosomes Transfer Assay

THLE-2 and CCD841-CoN recipient cells (1 × 10^4^) were seeded on 12-well plates and incubated for 8 days with SW480-derived exosomes (2 × 10^9^) in the presence of IL-6, TNFα, and IL1β (20 ng/mL each, Peprotech); culture media were completely replaced every other day. Twenty-four hours after the last treatment, cells were harvested and divided into two parts: one was seeded and cultured for 10 days without exosomes but in the presence of cytokines, and the other was used for DNA and RNA extraction using a QIAamp DNA Mini and RNeasy Kit (Qiagen), respectively. After 10 days of cytokine treatment, cells were harvested and nucleic acids were extracted.

To assess the presence of mutated KRAS c.35G > T and TP53 c.818G > A, DNA was analyzed by droplet digital PCR, as described above. Data were expressed as mutated copy number normalized by the amount of DNA.

For RNA analysis, the one step RT-ddPCR Advanced Kit and validated primer kits and probes were used. Thermal cycling conditions consisted of: 60 min at 43 °C (reverse transcription), 95 °C for 10 min, followed by 40 cycles at 94 °C for 30 s, 55 °C for 1 min, and 98 °C for 10 min (ramp rate 2 °C/s).

Discrimination between negative and positive droplets was achieved by manually setting a fluorescence amplitude threshold based on the signal from untreated cells.

### 4.7. Statistical Methods

Data are reported as mean ± standard deviation. Statistical analysis was performed using the GraphPad Prism software (version 7). Data were tested for normal distribution using the Kolmogorov–Smirnov test. Measures were analyzed using a one-way analysis of variance followed by the Bonferroni or Dunnett post-test. Values of *p* < 0.05 were considered significant.

## Figures and Tables

**Figure 1 ijms-22-06258-f001:**
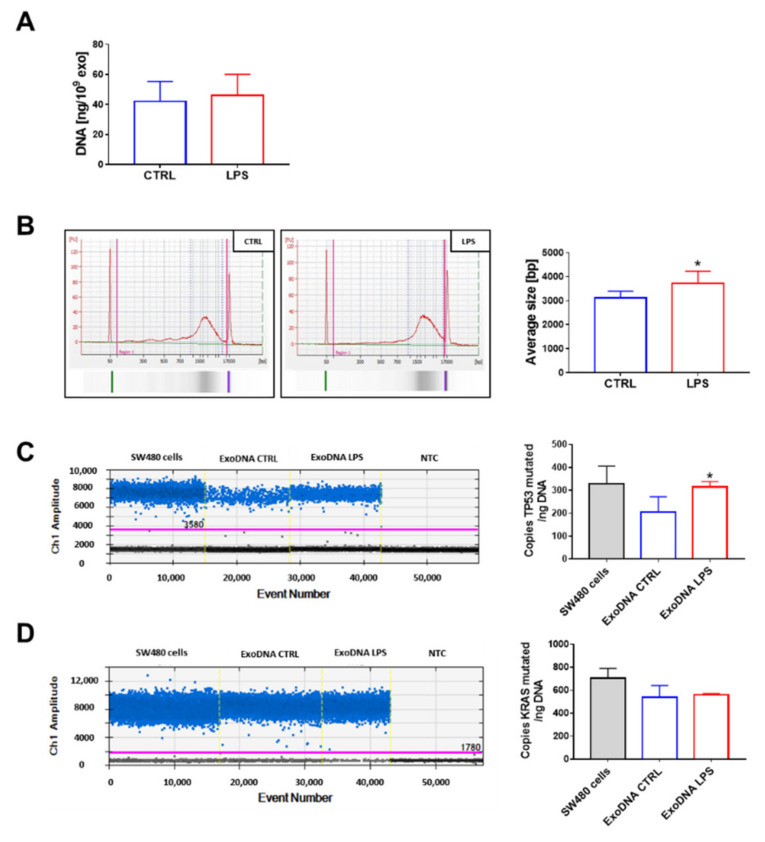
Fragments of dsDNA containing *TP53* c.818G > A and *KRAS* c.35G > T mutations were found in SW480-derived exosomes. Exosomal DNA was extracted from vesicles produced by untreated (ExoDNA CTRL) or lipopolysaccharides (LPS) stimulated (ExoDNA LPS) cells and analyzed using Qubit (**A**) and Bioanalyzer (**B**). The mutated copy analysis of *TP53* (**C**) and *KRAS* (**D**) was conducted using droplet digital PCR, and representative results are reported (intensity plot). NTC (negative control). Data are shown as mean (*n* = 6) ± SD. * *p* < 0.05, compared with the exosomes produced by untreated cells.

**Figure 2 ijms-22-06258-f002:**
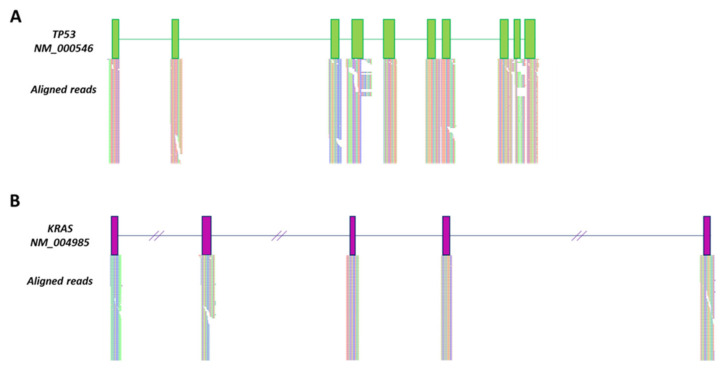
SW480-produced exosomes carried whole *TP53* and *KRAS* genes. Aligned whole-coding region reads of *TP53* (**A**) and *KRAS* (**B**). Green and purple boxes represent *TP53* and *KRAS* exons, respectively. Slanted lines represent *KRAS* intronic regions that have been cropped for graphical purposes.

**Figure 3 ijms-22-06258-f003:**
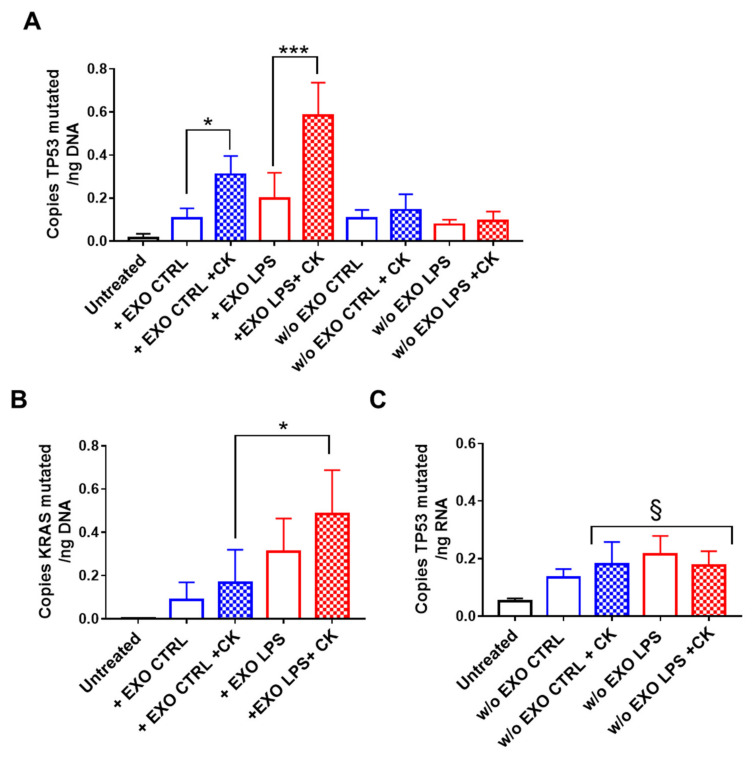
Exosomal-derived *TP53* c.818G > A was integrated into the DNA of normal colonic epithelial cells. CCD841-CoN cells were incubated with exosomes released from SW480 cells (untreated: EXO CTRL, LPS-stimulated: EXO LPS) in the presence of pro-inflammatory cytokines (CK) for 8 days and then cultured for an additional 10 days without exosomes. The presence of mutated *TP53* (**A**,**C**) and *KRAS* (**B**) were assessed by droplet digital PCR on extracted DNA (**A**,**B**) and RNA (**C**). Data are shown as mean (*n* = 4) ± SD. * *p* < 0.05, *** *p* < 0.001 compared with cells cultured in the absence of cytokines; ^§^
*p* < 0.05, compared with untreated cells.

**Figure 4 ijms-22-06258-f004:**
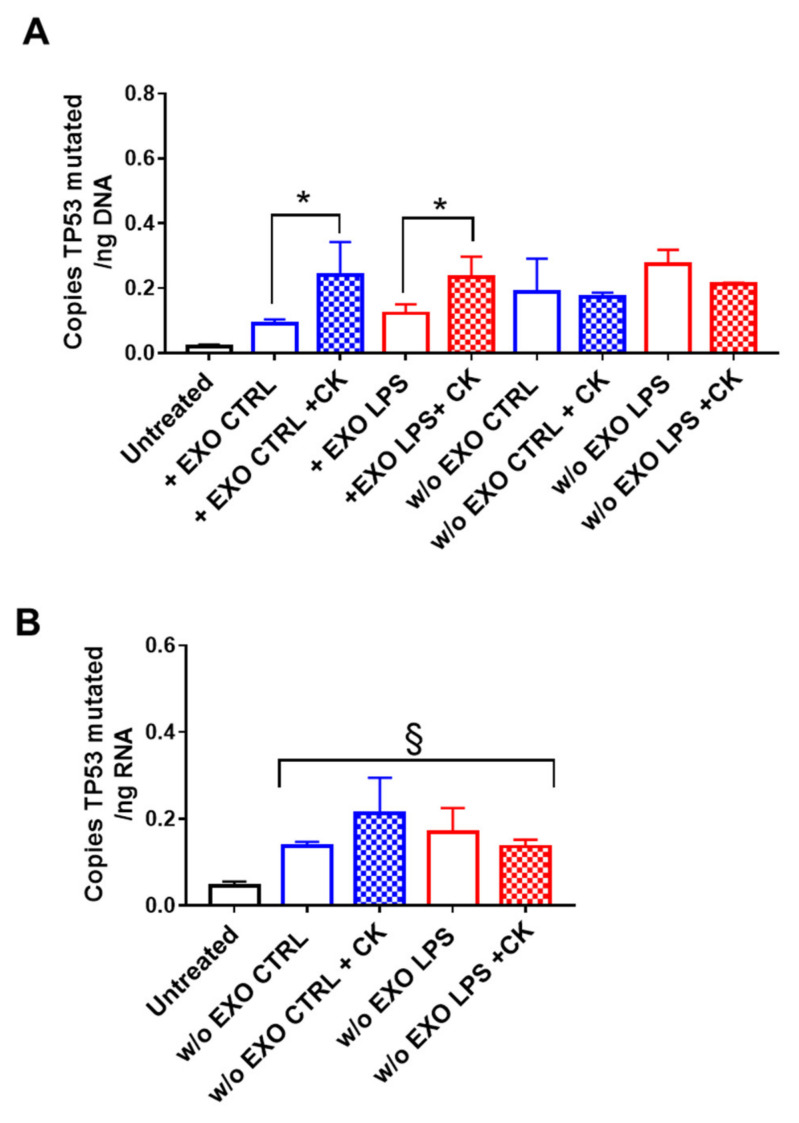
Exosome-derived *TP53* c.818G > A was integrated into the DNA of normal recipient hepatocytes. THLE-2 cells were incubated with exosomes released from SW480 cells (untreated: EXO CTRL, LPS stimulated: EXO LPS) in the presence of pro-inflammatory cytokines (CK) for 8 days and then for an additional 10 days without exosomes (**A**). Mutated *TP53* was detected by droplet digital PCR on extracted DNA (**A**) and RNA (**B**). Data are shown as mean (*n* = 4) ± SD. * *p* < 0.05, compared with cells cultured in the absence of cytokines; ^§^
*p* < 0.05, compared with untreated cells.

## Data Availability

Not applicable.

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
