# Peer review of "Pro-Inflammatory Microenvironment Modulates the Transfer of Mutated TP53 Mediated by Tumor Exosomes"

_ijms, 2021, doi:10.3390/ijms22126258_

Round 1
Reviewer 1 Report
This study shows that tumor-derived exosomes transfer dsDNA fragments horizontally to other cells, affecting the microenvironment. The authors demonstrated that SW480-derived exosomes effectively deliver dsDNA with the full coding sequence of mutated TP53 to epithelial and hepatic cells, and that oncogenic DNA delivery is enhanced in a pro-inflammatory environment.
Please find below my comments:
- In Figure 1, the authors showed that stimulation of SW480 cells with LPS had no effect on KRAS gene packaging in exosomes. However, the results of Figure 3 show that the LPS-stimulated SW480-derived exosome-treated group in CCD841-CoN cells significantly increased KRAS mutations compared to the untreated group. The author should discuss the reasons for these results.
- The authors should provide a clear rationale of exactly why Figure 3's A and C were presented separately. There seems no reason to perform in vivo experiments under different culture conditions (e.g. cultivation period) between the groups treated w/ and w/o exosomes. Please describe the meaning of acronyms indicated in the figures in the main manuscript or figure legend.
- There are many grammar issues or confusing statements. English should be checked by a native English speaker.
Author Response
Reviewer 1
This study shows that tumor-derived exosomes transfer dsDNA fragments horizontally to other cells, affecting the microenvironment. The authors demonstrated that SW480-derived exosomes effectively deliver dsDNA with the full coding sequence of mutated TP53 to epithelial and hepatic cells, and that oncogenic DNA delivery is enhanced in a pro-inflammatory environment.
Please find below my comments:
- In Figure 1, the authors showed that stimulation of SW480 cells with LPS had no effect on KRAS gene packaging in exosomes. However, the results of Figure 3 show that the LPS-stimulated SW480-derived exosome-treated group in CCD841-CoN cells significantly increased KRAS mutations compared to the untreated group. The author should discuss the reasons for these results.
We thank the reviewer for his/her observation. We found that treatment of SW480 cells with LPS does not increase the content of exosome-transported mutated KRAS. The increase of KRAS, which we observed in CCD841 recipient cells treated with exoLPS, could be the result of their more efficient internalization. In fact, it has already been reported that stimuli present in the microenvironment can influence not only the exosomal content but also the ability to internalize vesicles (Harmati, M et al. Sci. Rep. 2019, 9; Kothandan, V.K. Vaccines 2020, 8, 172) by recipient cells. In any case, although there is a higher amount of mutated KRAS in exoLPS-treated recipient cells, it is lost as soon as exosome treatment is stopped. Therefore, we added the following sentence in the text: "We hypothesize that LPS treatment of SW480 affects the composition of released exosomes, enriching their content with certain oncogenes (as is the case for TP53), and also promoting their receptor- and raft-mediated endocytosis by recipient cells (as is likely the case for KRAS in our experimental model)".
- The authors should provide a clear rationale of exactly why Figure 3's A and C were presented separately. There seems no reason to perform in vivo experiments under different culture conditions (e.g. cultivation period) between the groups treated w/ and w/o exosomes. Please describe the meaning of acronyms indicated in the figures in the main manuscript or figure legend.
The only reason we separated the graphs was to make them more understandable. As suggested by the reviewer, we merged the two figures, specifying the acronyms in the figure legend. Similarly, we also combined Figure 4 A and C.
- There are many grammar issues or confusing statements. English should be checked by a native English speaker.
As suggested by the reviewer, the entire manuscript has been reviewed for grammar errors and typos.
Reviewer 2 Report
This interesting paper reports on the discovery that exosomes derived from SW480 cells carry dsDNA fragments containing the entire coding sequence of TP53 and KRAS genes. The influence of different stimuli onto the packaging of mutated genes into these nanovesicles has been investigated. The stimuli tested included lipopolysaccharides (LPS) and pro-inflammatory cytokines, such as IL1β, IL6 and TNFα cytokines. The Authors have found that LPS promotes selectively packaging of TP53 genes but not the KRAS genes. Furthermore, an efficient horizontal transfer of mutated sequences into the recipient cells, such as the normal CCD841-CoN colon epithelial cells and THLE-2 hepatic cells, has been observed. It has been found that the pro-inflammatory stimulation promotes the horizontal transfer of oncogenes by exosomes.
I recommend the paper for publication after minor revision addressing the issues listed below:
- The abbreviations should be defined at their first use, e.g., LPS, CK, AE.
- The analysis of exosomes offers a great potential for clinical diagnostics and therapy since the important cancer biomarkers can be detected in exosomes without invasive biopsies. Recently, it has been shown that the anti-apoptotic survivin (Sur), is carried by exosomes excreted by cancer cells and transferred to distal healthy cells inducing metastasis, see for instance: Biosensors and Bioelectronics, 2019, 137, 58-71. This relevant literature reference should be cited.
- There are only few typographical and English errors which require corrections, e.g.:
Line 16: “stimulation … promote” – should be “stimulation … promotes”; also, the same in line 21;
Line 137: “cultured for other10 days” – better would be: “cultured for another 10 days”;
Line 299: “completed culture media was replaced” – should be: “completed culture media were replaced”;
Line 300: “dived” – should be “divided”.
Author Response
Reviewer 2
This interesting paper reports on the discovery that exosomes derived from SW480 cells carry dsDNA fragments containing the entire coding sequence of TP53 and KRAS genes. The influence of different stimuli onto the packaging of mutated genes into these nanovesicles has been investigated. The stimuli tested included lipopolysaccharides (LPS) and pro-inflammatory cytokines, such as IL1β, IL6 and TNFα cytokines. The Authors have found that LPS promotes selectively packaging of TP53 genes but not the KRAS genes. Furthermore, an efficient horizontal transfer of mutated sequences into the recipient cells, such as the normal CCD841-CoN colon epithelial cells and THLE-2 hepatic cells, has been observed. It has been found that the pro-inflammatory stimulation promotes the horizontal transfer of oncogenes by exosomes.
I recommend the paper for publication after minor revision addressing the issues listed below:
- The abbreviations should be defined at their first use, e.g., LPS, CK, AE.
As suggested by the reviewer, we have specified the acronyms in figure legends and text. For AE buffer, we could not indicate the meaning of the abbreviation because it is named in this way in the kit instructions; however, we have specified its composition.
- The analysis of exosomes offers a great potential for clinical diagnostics and therapy since the important cancer biomarkers can be detected in exosomes without invasive biopsies. Recently, it has been shown that the anti-apoptotic survivin (Sur), is carried by exosomes excreted by cancer cells and transferred to distal healthy cells inducing metastasis, see for instance: Biosensors and Bioelectronics, 2019, 137, 58-71. This relevant literature reference should be cited.
As suggested by the reviewer, we cited this important reference in the new version of the manuscript.
- There are only few typographical and English errors which require corrections, e.g.:
Line 16: “stimulation … promote” – should be “stimulation … promotes”; also, the same in line 21;
Line 137: “cultured for other10 days” – better would be: “cultured for another 10 days”;
Line 299: “completed culture media was replaced” – should be: “completed culture media were replaced”;
Line 300: “dived” – should be “divided”.
We thank the reviewer for his/her corrections. We modified the text accordingly.
Reviewer 3 Report
Proinflammatory microenvironment modulates the transfer of mutated TP53 mediated by tumor exosomes.
By Rossana Domenis et al.
The overarching theme of this manuscript is that tumor derived exosomes in a pro-inflammatory micro-environment, are capable of transmitting mutated TP53 gene and somehow promote its integration into the genome of the recipient cells. In other words the authors are proposing that tumor exosomes can transform normal recipient cells. Such ideas have been suggested before without hard experimental evidence. Hence this is the discovery idea and novel finding. Their experimental approach was simple and easy to follow. For rigor and reproducibility of the data, I would suggest the following.
- Repeat experiments represented in Figs 3 and 4 and after initial treatment with exosomes in the presence and absence of inflammatory cytokines, allow the cells to grow for 20 days without tumor derived exosomes but with frequent passaging. At the end of the 20 days search for mutated TP53 to indicate whether integration occurred. It is important to rigorously prove this point otherwise the mechanistic underpinning of their studies will be weak.
- One other point the authors need to consider is that since they used a polymer precipitation method to isolate their exosomes, they could not be sure that the DNA was transferred by exosome which are now regarded as small extracellular vesicles as opposed to large extracellular vesicles sometimes referred to as oncosomes. This possibility need to be discussed because more and more studies now question whether or not double stranded DNA is packaged in exosomes and not oncosomes.
Apart from those two points I have raised, the language of the manuscript needs thorough editing. The following are some queries and suggestions related to overall presentation of the manuscript.
- Line 13 should read …the effect of DNA packaging and transfer into recipient cells have yet to be investigated and not…transfer into recipient cells have not been yet investigated.
- Line 31 should read…. Exosomes bear specific repertoires of nucleic acids and proteins… and not ….repertoires of acid nucleic and proteins.
- Line 51 should read ..genotype is yet to be clarified and not ..genotype, it has not been yet clarified.
- In line 58 did they mean …host cells [20] or …recipient cells [20]?
Author Response
Reviewer 3
The overarching theme of this manuscript is that tumor derived exosomes in a pro-inflammatory micro-environment, are capable of transmitting mutated TP53 gene and somehow promote its integration into the genome of the recipient cells. In other words the authors are proposing that tumor exosomes can transform normal recipient cells. Such ideas have been suggested before without hard experimental evidence. Hence this is the discovery idea and novel finding. Their experimental approach was simple and easy to follow. For rigor and reproducibility of the data, I would suggest the following.
Repeat experiments represented in Figs 3 and 4 and after initial treatment with exosomes in the presence and absence of inflammatory cytokines, allow the cells to grow for 20 days without tumor derived exosomes but with frequent passaging. At the end of the 20 days search for mutated TP53 to indicate whether integration occurred. It is important to rigorously prove this point otherwise the mechanistic underpinning of their studies will be weak.
We agree with the reviewer on the importance of doubtlessly prove DNA integration. We have already tried to culture treated cells without exosomes for up to 30 days. The results showed no differences in the amount of TP53 mutated copies compared to cells analysed after 10 days culture. We specified our data in the results section:“ The presence of mutated TP53 was detected up to 30 days without exosome treatments (data not shown), and because no significant differences were observed compared with the analysis after 10 days, the experiment was not extended beyond that.”
One other point the authors need to consider is that since they used a polymer precipitation method to isolate their exosomes, they could not be sure that the DNA was transferred by exosome which are now regarded as small extracellular vesicles as opposed to large extracellular vesicles sometimes referred to as oncosomes. This possibility need to be discussed because more and more studies now question whether or not double stranded DNA is packaged in exosomes and not oncosomes.
We thank the reviewer for his/her observation. We know that large oncosomes could be involved in the transfer of DNA. However, before proceeding polymer precipitation, cells supernatants were filtered through a 0.2 μm filter to remove particles larger than 200 nm. Oncosomes are in a size range from 1 μm up to 10 μm, then it is plausible to assume that they are not present in our exosomal fraction.
Apart from those two points I have raised, the language of the manuscript needs thorough editing. The following are some queries and suggestions related to overall presentation of the manuscript.
Line 13 should read …the effect of DNA packaging and transfer into recipient cells have yet to be investigated and not…transfer into recipient cells have not been yet investigated.
Line 31 should read…. Exosomes bear specific repertoires of nucleic acids and proteins… and not ….repertoires of acid nucleic and proteins.
Line 51 should read ..genotype is yet to be clarified and not ..genotype, it has not been yet clarified.
In line 58 did they mean …host cells [20] or …recipient cells [20]?
As suggested by the reviewer, the entire manuscript has been reviewed for grammar errors and typos.
Round 2
Reviewer 1 Report
It seems that the authors revised the manuscript as per the comments.
Reviewer 3 Report
The authors have addressed all the questions I asked. I am now satisfied with the presentation of the manuscript